# The Other Side of the "League of Stars": Analysis of the Financial Situation of Spanish Football

Rudemarlyn Urdaneta-Camacho [1], Juan Carlos Guevara-Pérez [1,2,*], Emilio Martín Vallespín [1,*]
and Néstor Le Clech [3]

1   Faculty of Economics and Business, University of Zaragoza, 50005 Zaragoza, Spain
2   IGOID Research Group, Department of Physical Activity and Sport Sciences, University of
    Castilla-La Mancha, 45071 Toledo, Spain
3   Department of Economics and Business, National University of Quilmes, Roque Sáenz Peña 352,
    Bernal B1876, Buenos Aires, Argentina
*   Correspondence: jguevara@unizar.es (J.C.G.-P.); emartin@unizar.es (E.M.V.)

**Abstract:** This paper analyses the effectiveness of the financial control system implemented by the Spanish professional football League in 2015 as a tool to improve the governance of clubs in the first and second divisions as well as its probable impact on competition. Classic financial ratios are used to analyse the financial situation of the clubs both before and after the implementation of the Regulation, as well as during the first season affected by the COVID-19 pandemic. Next, the Herfindahl index is calculated to measure the concentration in the distribution of the main funding sources and is incorporated as a dependent variable in a regression model. Although a cause–effect relationship is not certain, the results suggest that the economic control measures imposed by LFP have contributed to improving the financial situation of Spanish football in the short term, but may promote imbalances between clubs that undermine the sustainability of the current management model and, therefore, of the competition system. Unlike in other sectors, the football business requires more competition to maximise profits. In this context, it would be advisable to reach agreements between clubs to weaken the bargaining position of footballers. The paper shows the effect of the intervention of a regulatory body, in this case, LFP, in the functioning of a competitive market.

**Keywords:** financial fair play; financial sustainability; football clubs; COVID-19

## 1. Introduction

Beyond the sporting spectacle and the passion it generates among fans every time the ball is kicked, football in Spain is a social phenomenon that attracts an astonishing amount of media attention, generating a huge amount of money that seems to have no limit. To guarantee the medium and long-term viability of football clubs, this business must be managed professionally and efficiently to try to achieve the necessary alignment of sporting and economic results. On the pitch, Spanish football has enjoyed its best period of sporting success over the last decade, both at the national team level, with the 2008 and 2012 European Championships and the 2010 World Cup, and at the club level. Some of the best world football stars compete in the Spanish league and their clubs usually occupy the top positions in European competitions as well as achieving the highest coefficient rankings of the Union of European Football Associations (UEFA).

However, the situation is very different in the economic sphere. The compulsory transformation of the clubs into sport-limited companies (except for Real Madrid, F.C. Barcelona, Ath. Bilbao and Osasuna) after the implementation of Sports Law in the early 1990s did not achieve the desired effects in terms of financial stability. During the following two decades, Spanish football continued to accumulate debts and losses up to the point of calling into question the sustainability of the business model and the risk of adulteration of the competition by financial doping.

While it is true that the same financial problems affect almost the entire European football industry (Barajas and Rodríguez 2010), Spain is by far one of the main countries affected, with an important number of clubs having serious difficulties in paying players, workers, suppliers and the Tax Agency itself. Proof of this is that more than 20 clubs (first or second division) declared bankruptcy between 2004 and 2012. In some cases, forcing their disappearance and/or refoundation. In addition, a growing number of foreign investors joined the shareholding of historic clubs such as Atlético de Madrid, Español, Granada, Málaga, Valencia and more recently Real Zaragoza (Mora Zaya 2021). To address this situation, football's main governing bodies, UEFA and Professional Football League (Liga de Futbol Profesional (LFP)), were forced to take initiatives aimed at improving football's financial viability by increasing revenues from TV contracts and controlling spending.

A peculiarity of Spanish football is that they maintain close links with the public administrations around them; although, the clubs are run by private entities. The politicians know that the degree of loyalty of the fans towards their teams is much greater than that of the customer of any other type of business, including the voters of a political party (Hassan and Hamil 2010; Michie and Oughton 2005). Thereby, politicians try to attract some of this fidelity in their favour. This has given rise to a groundswell of opinion critical of the privileges granted to the football world, and there are calls for greater control and transparency of clubs so that they are subject to the same business rules as the other organisations.

This study proposes to carry out an analysis of the financial sustainability of Spanish professional football's first-division and second-division through the analysis of solvency, profitability, indebtedness and liquidity ratios, which are subsequently used to explain the concentration of clubs' income obtained through the Herfindahl index (H-index) in a context in which the main national and international competition governing bodies (La Liga and UEFA) have implemented economic control regulations that seek to strengthen the economic viability of the clubs, as well as avoiding adulterations of the competition derived from "financial doping". In this way, the results of this study will help to better understand the economic particularities of the Spanish football industry compared to typical references in other sectors.

One of the main contributions of this study is to carry out a dynamic analysis of the financial situation of Spanish clubs that complements the traditional Consejo Superior de Deportes (CSD) and LFP reports that focus on the overall situation of the leagues at a certain time based on averages where the financial situation of a few clubs affects the vast majority. This analysis is particularly timely in a period of uncertainty such as the current one, caused by the COVID-19 pandemic, which has led to a severe reduction in revenues over the last two seasons and uncertain prospects for the coming years. The final objective is to evaluate the effectiveness of UEFA and LFP's economic control measures to turn Spanish professional football clubs into financially viable organisations[1].

In order to meet the proposed objectives, the rest of the paper is organised as follows: Section 2 makes a brief presentation on the economic context of football in Europe and on the main regulations that constitute the regulatory framework of Spanish football, both from an accounting and financial point of view. Sections 3 and 4 describe the methodology to be used and discuss the results obtained from its application. Finally, Section 6 presents the main conclusions of the work.

## 2. Literature Review

Despite the fact that the European football industry has been perceived as a highly regulated and organised sector (Foster 2000), the truth is that European football clubs and leagues have been characterised by chronic financial instability derived in part from the prioritisation of sporting results over economic ones (Garcia-del-Barrio and Szymanski 2009). Thus, we can find a wide range of studies documenting the insolvency, lack of policies and poor budget execution of football clubs, in addition to some fraudulent behaviour (Barros 2006; Lago et al. 2006; Panagiotis 2011). Additionally, the financial crisis of 2008

affected negatively the revenue-generating capacity of European clubs, which, according to the UEFA Executive Committee (UEFA 2012), put the future viability of many of them at risk.

Intending to improve the financial health of clubs, the governing bodies of the competitions decided to establish measures for the financial control of clubs (Dimitropoulos 2011). These regulations try to avoid possible adulterations in the outcome of competitions due to a mismatch between the expenditure of some teams and the revenue they generate. Thus, in 2010, UEFA adopted the regulation known as "financial fair play (FFP)" (UEFA 2012), which monitors the financial information published in the annual accounts of clubs participating in European competitions and places limits on debts, expenses and sponsorship. So that those clubs that deviate from the financial discipline required by UEFA would be subject to significant sanctions that could even result in their exclusion from European competitions, as has been the case for A.C. Milan, one of the clubs with the best European record, during the 2019–20 season. In turn, the introduction of FFP is an element that may lead to the introduction of creative accounting practices by clubs to accommodate the financial information presented in their financial statements to UEFA's requirements (Dimitropoulos et al. 2016).

European football clubs seem to be more win maximisers than profit maximisers (Dimitropoulos and Tsagkanos 2012). Therefore, the UEFA's FFP can be considered the first regulation of European football that tries to transform the traditional approach of "utility maximisation" (UM), in which victories are prioritised, into a "profit maximisation" (PM) strategy, which is the most common approach in the Anglo-Saxon world (Nicoliello and Zampatti 2016).

Since UEFA's FFP only affects clubs participating in European competitions, it is necessary for each country to establish its own control mechanisms (for example, Dermit-Richard et al. 2019), to stimulate the financial discipline of the majority of professional clubs without the expectation of participating in such European tournaments. In Spain, with the implementation of Law 10/1990 of 15 October 1990 on Sport, the professional leagues were granted financial control and supervision over their members.

In 2014, the LFP introduced the "Regulations for the economic control of Clubs and Sports Corporations (Sociedades Anónimas Deportivas, (SAD)) affiliated to the LFP" Economic Control Regulation (Reglamento de Control Económico (RCE)), which establishes rules for supervision and control, in line with the objectives of UEFA's FFP to improve the economic and financial capacity of clubs by increasing their transparency and credibility, fostering greater discipline and rationality in finances by encouraging clubs to operate based on their revenue capacities, and promoting responsible spending to guarantee the viability and sustainability of competition and sporting entities (LFP 2016; UEFA 2012).

The finances of European football have experienced improvements in recent years as a result of the new regulations (Franck 2018), reducing their losses, especially in Spain (Ahtiainen and Jarva 2020; Sánchez et al. 2019), where the LFP's RCE has joined in improving the economic situation of the clubs (Calahorro-López et al. 2022; Urdaneta et al. 2021). In 2020, the professional competition industry showed positive results for the eighth consecutive year (CSD 2020). To do this, many clubs have had to adopt strict rationalisation measures at the cost of limiting the sporting potential of their teams (Plumley et al. 2019).

On the financing side, in recent years different initiatives have also been implemented, both individually and collectively, to increase the revenues of Spanish football. Thus, since the 2016/17 season, the LIGA has been negotiating the sale of TV rights centrally, which has led to a significant increase in gross revenues compared to the previous system of individual negotiation[2]. In addition, 38 out of the 42 LFP clubs[3] have recently signed the sale of 10% of the LFP business to the Luxembourg fund CVC Capital Partners for 50 years for around 2.1 billion euros.

Currently, European clubs are also preparing to cope with the economic consequences of COVID-19, which according to a recent study by (KPMG 2020), could cost the five major

European leagues around EUR 4 billion, mainly due to lower TV and commercial revenues (KPMG 2020).

### 3. Methodology

In order to study the financial situation of the clubs, this study analyses basic ratios of liquidity, solvency, debt and profitability just before and after the implementation of Spanish LPF's RCE regulations, and during the first season of the COVID-19 pandemic.

Liquidity analysis allows us to assess the ability of clubs to meet their short-term debts with their short-term available resources, i.e., without the need to divest themselves of long-term investments, which would lead to a gradual decapitalisation of the institution. Although there are other more restrictive magnitudes to evaluate liquidity such as the acid test or the immediate liquidity ratio, we have preferred to use the general current assets/current liabilities ratio since this is the magnitude used to calculate the working capital. Moreover, football clubs do not usually have inventories so the results hardly differ from those calculated with the acid test.

Solvency focuses on the ability of an institution to meet all its debts within its available resources in both the short and long term, i.e., regardless of the institution's continuity. The debt ratio, on the other hand, relates the use of external and own sources of financing in such a way that the higher the debt ratio is, the higher the risk for both creditors and shareholders, due to the leverage effect.

Finally, the economic profitability ratio attempts to measure the productivity of the activity, relating the results generated by the activity to the resources used, i.e., to the assets. When the economic profitability is higher than the average cost of debt, indebtedness plays a positive role since the institution can obtain higher income from the financing raised than the income via interest. The opposite is true if the economic return is lower than the cost of debt.

In addition, we have preferred to use ROA instead of ROE, because some clubs have negative equity, which distorts the evaluation and interpretation of ROE for those entities with equity close to 0 (Sánchez et al. 2016). Furthermore, the calculation of ROA has been defined by the ratio EBIT/Total Assets. We have used EBIT (Earning Before Interest and Taxes) in the numerator instead of EBI, which is commonly used to avoid distortions caused by changes in taxation during the period of analysis. For example, in 2015, the general corporate tax rate decreased from 30% to 25%. This reduction in the tax rate would lead to a partial improvement of the EBI, which is not attributable to the management of the football clubs.

On the other hand, it is important to be aware of the standards by which the accounting information is constructed, as well as the particularities of the sector being analysed. In this case, it should be borne in mind that most of the assets are valued according to the historical cost criterion, which will always be equal to or less than the market value, and that not all the elements that provide value in a football club are included in its annual accounts, such as the value of the sports staff.

Additionally, the composition of the clubs' main sources of income is analysed by using the H-index, which measures the degree of diversification of income and, consequently, their financial dependence. The H-index is a frequently used indicator in different fields as a measure of concentration and could therefore be considered a good approximation to assess the degree of financial dependence of clubs on some source of income. For this purpose, revenues will be classified into four different types depending on their origin: (1) TV broadcasting revenues, (2) Advertising and marketing revenues, (3) Ticket and membership revenues (Match day), (4) Other revenues (player transfers, subsidies, etc.). On this occasion, the H-index will range between 0.25 and 1, so that a value of H = 1 indicates maximum concentration in revenues and 0.25 maximum possible dispersion and, therefore, less financial dependence.

The mathematical formula for the index is as follows:

$$H - index = \sum_{i=1}^{m} p_i^2 \qquad (1)$$

where "H-index": is Herfindahl Index, "i": is each of the 4 main funding sources of the football clubs and "$p_i$": is the percentage of the revenue in each club represented by the funding source "i".

The population under study consists of 1stD and 2ndD clubs that participated in the 2013/14 (before FFP and RCE), 2016/17 (after FFP and RCE) and 2019/20 (during the COVID-19 pandemic) seasons. The sample was reduced to the 25 teams that submitted economic–financial information in the three seasons analysed. As can be seen in Table 1, the sample is very heterogeneous in all the financial parameters and includes clubs with revenues of around 500 million along with others that do not reach 1 million. Of all of them, the only entities that have not converted to SAD are Barcelona, Madrid, Athletic Club and Club Atlético Osasuna as a consequence of the fact that in 1990, the Sports Limited Companies Act was passed, which dictated that those with a positive net worth balance since the 1985/86 season should not convert to SAD.

**Table 1.** Descriptive statistics of some of the most representative financial statement items.

| Variable | Season | *n* | Minimum | Maximum | Mean | Standard Deviation |
|---|---|---|---|---|---|---|
| Current Assets | 2013–2014 | 25 | 220,209.0 | 271,002,000.0 | 37,420,035.3 | 64,337,014.7 |
| | 2016–2017 | 25 | 1,504,451.3 | 325,946,105.1 | 59,087,235.5 | 91,259,706.5 |
| | 2019–2020 | 25 | 3,352,510.9 | 368,522,000.0 | 71,435,128.1 | 108,075,962.0 |
| Current Liabilities | 2013–2014 | 25 | 210,346.0 | 378,766,000.0 | 71,652,831.8 | 119,626,773.9 |
| | 2016–2017 | 25 | 2,104,880.0 | 587,869,000.0 | 96,755,918.7 | 172,162,120.9 |
| | 2019–2020 | 25 | 2,335,156.7 | 970,349,000.0 | 118,614,675.0 | 219,708,133.9 |
| Total Assets | 2013–2014 | 25 | 4,080,495.0 | 977,486,000.0 | 145,180,165.4 | 230,116,794.5 |
| | 2016–2017 | 25 | 5,896,336.9 | 1,074,662,000.0 | 196,352,449.2 | 287,805,721.4 |
| | 2019–2020 | 25 | 7,742,860.0 | 1,474,027,000.0 | 270,801,609.5 | 421,871,495.8 |
| Heritage | 2013–2014 | 25 | −97,644,447.7 | 369,395,000.0 | 23,340,252.1 | 79,460,584.0 |
| | 2016–2017 | 25 | −77,116,807.0 | 463,476,000.0 | 48,725,261.5 | 96,563,158.0 |
| | 2019–2020 | 25 | −73,400,815.0 | 532,925,000.0 | 63,843,364.7 | 107,765,760.9 |
| Total Liabilities | 2013–2014 | 25 | 591,087.0 | 608,091,000.0 | 121,839,913.2 | 171,944,997.5 |
| | 2016–2017 | 25 | 2,541,174.1 | 847,195,874.1 | 147,627,187.8 | 224,955,715.8 |
| | 2019–2020 | 25 | 4,008,921.7 | 1,438,840,000.0 | 206,958,204.9 | 363,671,250.7 |
| Revenues | 2013–2014 | 25 | 460,910.0 | 500,505,003.0 | 66,836,228.6 | 126,710,685.8 |
| | 2016–2017 | 25 | 1,766,863.5 | 671,864,000.0 | 105,541,958.6 | 166,616,529.7 |
| | 2019–2020 | 25 | 1,580,412.3 | 708,257,000.0 | 116,425,043.8 | 188.648.078,9 |
| Operating Profit | 2013–2014 | 25 | −4,931,813.5 | 57,997,000.0 | 9,526,515.4 | 15.257.188,9 |
| | 2016–2017 | 25 | −4,716,356.5 | 30,993,000.0 | 11,245,179.3 | 12.587.226,1 |
| | 2019–2020 | 25 | −99,795,000.0 | 46,949,955.1 | 1,763,973.9 | 24,745,633.7 |

The financial data have been manually extracted from the transparency portals contained in the websites of the Spanish professional teams and also through the SABI database. This fact constitutes a contribution of the present study compared to previous studies limited to the information obtained from databases, since, by relying directly on the financial statements of the clubs, it was possible to analyse in more detail items that have traditionally been considered generically as total income, the breakdown of which has made it possible to determine the concentration of its components through the H-index, or contrast the detail of the proportion of the expenditure of player wages to determine the thresholds provided for by regulations.

Finally, through a panel of data econometric analyses, the study has also established the influence of the financial variables and the economic control regulation on financial dependence expressed as income concentration through the Herfindahl index. These relationships are expressed in the following equation:

$$H - index = f(\text{Liquidity}; \text{Solvency}; \text{Indebtedness}; \text{ROA}; \text{Revenue}; \text{F. Control}) \quad (2)$$

where H-index is the Herfindahl index; Liquidity is the current assets over current liabilities ratio; Solvency is the total assets over total liabilities ratio, ROA is the ratio EBIT/Total Assets; Revenue is the total income of each club in millions; and F. Control is a dummy variable that takes value 0 before the 2015/16 season and 1 otherwise.

In order to choose the adequate econometric technique, we must take into account the bounded nature of our dependent variable, which has a range of values between 0 and 1. This characteristic makes OLS or other similar techniques not appropriate. In this case, the beta regression model (BRM) is a suitable alternative. This technique implements maximum likelihood estimators based on the beta distribution. We use the proposal of Ferrari and Cribari-Neto (2004)[4].

One advantage of BRM is that no matter the functional form chosen, both models are consistent. Moreover, when the model is correctly specified, these techniques are also efficient. In other words, if the correct functional form is selected the estimations will be consistent and efficient.

The model fits a regression for the mean of the dependent variable, which is conditional to independent variables. There are distinct options to specify the link function for the conditional mean. For example, it also allows 'logit', 'probit', 'log-log' and 'complementary log-log'. In addition, the link function for the conditional scale must be specified for the BRM. The options are 'log', 'root' or 'identity', (for further details see Ferrari and Cribari-Neto (2004) and Smithson and Verkuilen (2006)).

To identify the appropriate functional form, we have followed Smithson and Verkuilen (2006), who recommend selecting the model that minimises the Bayesian Information Criterion (BIC). This procedure indicates that 'complementary log-log' is the most appropriate functional form. The option chosen for the conditional scale was 'log' since it is the most common and there were no differences with regard to the other two options.

## 4. Results

### 4.1. Analysis of the Effect of the Economic Control Regulation (2013/14–2016/17)

Table 2 shows the descriptive statistics of the main financial ratios of the football clubs calculated from the equity masses extracted from their financial statements.

**Table 2.** Descriptive statistics of main financial ratios.

| Variable | Seasons | *n* | Minimum | Maximum | Mean | Median | Standard Deviation |
|---|---|---|---|---|---|---|---|
| Liquidity | 2013–2014 | 25 | 0.183 | 3.957 | 0.813 | 0.615 | 0.785 |
|  | 2016–2017 | 25 | 0.115 | 1.629 | 0.761 | 0.691 | 0.397 |
|  | 2019–2020 | 25 | 0.145 | 4.13 | 0.958 | 0.766 | 0.873 |
| Solvency | 2013–2014 | 25 | 0.176 | 6.903 | 1.581 | 1.141 | 1.43 |
|  | 2016–2017 | 25 | 0.401 | 2.677 | 1.478 | 1.466 | 0.492 |
|  | 2019–2020 | 25 | 0.258 | 4.099 | 1.769 | 1777 | 0.814 |
| Indebtedness | 2013–2014 | 25 | −18.968 | 39.632 | 5.827 | 3.332 | 11.439 |
|  | 2016–2017 | 25 | −14.186 | 75.842 | 7.808 | 1.937 | 19.285 |
|  | 2019–2020 | 25 | −1.349 | 40.891 | 4.507 | 1.282 | 8.72 |
| ROA | 2013–2014 | 25 | −0.146 | 0.365 | 0.082 | 0.052 | 0.124 |
|  | 2016–2017 | 25 | −0.129 | 0.503 | 0.104 | 0.051 | 0.146 |
|  | 2019–2020 | 25 | −0.203 | 0.376 | 0.066 | 0.043 | 0.128 |

Regarding liquidity, which is the ratio of current assets to current liabilities, the results show a significant gap between the minimum and maximum values, which highlights the heterogeneous situation of the clubs. Although on average, liquidity is relatively close to unity, which is the value usually taken as a reference, the median is always below the average suggesting a compromised situation for most teams to honour their commitments in the short term. It is also worth noting the relative persistence in the behaviour of the clubs so that most of them show ratios without significant fluctuation between periods regardless of the sporting ranking achieved each season. This situation may be a consequence of the short-term indebtedness assumed by some teams to develop improvements to their infrastructures (CSD 2020) and forces most clubs to sign credit lines that are renewed year after year. When this is not enough, the clubs usually sell some players to balance their accounts.

When analysing solvency, measured as a ratio of total assets to total liabilities, the results show an improvement over the period analysed in terms of the median of the sample, suggesting that, at least in this area, the LFP's CER is achieving good results. Nonetheless, if we take a value of 1.5 as a benchmark, which is commonly used in some industries as an indicator of the ability of companies to meet their long-term commitments and the optimal balance between equity and debt, most clubs are still below this value in 2017 (see Table 2). However, in the football industry, it is very difficult to establish benchmark values to carry out comparisons of solvency. This is due to the nature of their main economic asset, namely, the playing squad, which is an intangible accounting asset and is only possible when a transfer fee has been paid for the player. Consequently, home-grown players and all those whose signings have already been amortised for accounting purposes will not appear on the clubs' balance sheets, even though they are the most frequently used asset to obtain a high amount of money through their transfer.

With regard to the debt ratio (ratio of liabilities to equity), the results show a fairly high average, which reflects an accentuated dependence on external financing and a great heterogeneity in the situation of the clubs, ranging from extraordinarily high ratios in clubs such as Real Zaragoza, Real Valladolid or Atlético de Madrid itself, to other clubs that present a less compromised situation, such as Ath Bilbao or Celta de Vigo (see Appendix A). Sometimes, this ratio reaches negative values, which occurs when the net worth of the entity is negative. This is a situation that has traditionally been referred to as technical bankruptcy. However, it can be seen that after the implementation of the LFP's RCE, the number of clubs with negative equity was reduced from 5 to 2. Thereby, the LFP's priority of protecting the long-term viability and sustainability of Leagues and Clubs is paying off. On the other hand, when a club goes from having a negative to a positive net worth, the debt ratio skyrockets even though it has improved its situation, as the denominator of the ratio is very low (see Real Zaragoza or Valladolid in Appendix A). This is the reason why the average indebtedness figure shown in Table 2 in 2017 is higher than in 2014 when the reality in 2019 showed that the 25 teams in the sample have decreased their debt ratio (see Appendix A).

Economic profitability is measured as earnings before interest and taxes divided by total assets. In general, positive profitability is observed in most cases, as only five teams had economic losses in the 2013/14 season and three in the 2016/17 season. Although, as for the other magnitudes, there is no reference indicator of financial profitability, we observe that the most prominent clubs at a sporting level, such as Real Madrid, Barcelona or Atlético de Madrid, show ratios that oscillate within a range between 2% and 4%, while other clubs show significant fluctuations from one season to the next.

On average, the economic profitability of the sample increased from 8% to 10% as a consequence of the implementation of FFP and the RCE. This improvement in profitability is probably due to the increase in revenues derived from the centralised sale of TV rights from the 2016/17 season onwards. The higher volume of revenue would also justify the increase observed in the standard deviation of ROA; although, the new distribution criterion mainly favours the smaller clubs. Thus, the largest clubs such as Real Madrid

even show decreases in ROA, while more modest clubs such as Real Zaragoza and Osasuna have improved their profitability, probably as a result of greater control by the LFP.

Appendix B shows the distribution of the clubs' revenues used to analyse the degree of dependence on a certain source using the Herfindahl index. For this purpose, following a criterion similar to that used by the Professional Football League, the clubs' revenues are classified under four headings: broadcasting revenues, advertising and marketing revenues, ticket and membership revenues (matchday) and other revenues.

The sale of television rights is the main source of revenue in Spanish football, representing on average 63% of the total revenue of the teams in the sample for the 2016–2017 season (see Table 3) and an item that has been gradually increasing in both amount and relative importance. This is largely due to the implementation of Real Decreto-ley 5 (2015) that centralised the sale of television rights. This allowed revenues to rise from 826 million euros in the 2013/14 season to more than 1.4 billion euros in the 2016/17 season (CSD 2020). Again, it is worth noting, the great heterogeneity in the situation of the clubs regards the percentage that this item represents, ranging from less than 38% of the revenues of Real Madrid, F.C. Barcelona and Atlético de Madrid, to more than 80% in the case of Osasuna, Huesca, Leganés and Eibar. The median is 69%, indicating that most clubs are concentrated in an above-average situation.

**Table 3.** Descriptive statistics: incomes and player wages.

| Variable | Seasons | *n* | Minimum | Maximum | Mean | Median | St. Dev. |
|---|---|---|---|---|---|---|---|
| TV Rights | 2013/14 | 19 | 0.282 | 0.777 | 0.390 | 0.375 | 0.269 |
| | 2016/17 | 24 | 0.033 | 0.897 | 0.626 | 0.685 | 0.217 |
| | 2019/20 | 25 | 0.034 | 0.892 | 0.603 | 0.643 | 0.222 |
| Advertising and Marketing | 2013/14 | 19 | 0.064 | 0.366 | 0.143 | 0.158 | 0.103 |
| | 2016/17 | 24 | 0.064 | 0.459 | 0.169 | 0.131 | 0.122 |
| | 2019/20 | 25 | 0.048 | 0.460 | 0.200 | 0.148 | 0.126 |
| Matchday | 2013/14 | 19 | 0.063 | 0.382 | 0.132 | 0.117 | 0.110 |
| | 2016/17 | 24 | 0.022 | 0.259 | 0.115 | 0.102 | 0.068 |
| | 2019/20 | 25 | 0.017 | 0.485 | 0.123 | 0.097 | 0.107 |
| Other Income | 2013/14 | 18 | 0.019 | 0.419 | 0.335 | 0.118 | 0.396 |
| | 2016/17 | 23 | 0.018 | 0.310 | 0.090 | 0.040 | 0.096 |
| | 2019/20 | 23 | 0.008 | 0.363 | 0.074 | 0.024 | 0.100 |
| Player Wages | 2013/14 | 25 | 41.282 | 245.172 | 73.013 | 63.414 | 41.147 |
| | 2016/17 | 25 | 25.231 | 105.223 | 54.786 | 55.347 | 18.447 |
| | 2019/20 | 25 | 35.612 | 93.156 | 63.045 | 59.352 | 14.994 |

Source: Author's elaboration.

Advertising and marketing revenues are the second most important source of income, representing on average 17% of each club's revenues, ranging from 4% for Ath. Bilbao to 45% for Barcelona. In this case, the median is around 13% of revenues, indicating that most clubs are concentrated below average.

Although income from ticket sales, season ticket holders and competitions (usually referred to as Matchday income), is the most basic source of funding for clubs (contributions from fans for consuming the live service in the stadium), in practice, it represents, on average, only 12% of the income for the season 2016–2017. Eibar is the club with the lowest share of this item in its revenues (2%) while at Albacete this source represents 26% of revenues.

The heading of "other income" corresponds to the rest of the income that does not derive from the club's corporate purpose but is complementary to it: operating subsidies, leases, compensation, sale of players, and so forth. These represent, on average, 9% of the clubs' total income, constituting the most volatile source of financing in the League. In the 2016/2017 financial year, they have been reduced by 25%, which makes sense if we take into account the redirection of the accounts by the teams with the implementation of

Real Decreto-ley 5 (2015), where the teams previously included the increase in television revenue in this item (see Appendix B).

As can be seen in Table 4, the Herfindahl index shows the high concentration of the clubs' income, indicating their dependence on one source.

**Table 4.** Herfindahl index.

| ID | Teams | Herfindahl Index | | |
|---|---|---|---|---|
| | | **2013–2014** | **2016–2017** | **2019–2020** |
| 1 | Atlético Madrid | 0.30 | 0.30 | 0.28 |
| 2 | Barcelona | 0.30 | 0.33 | 0.35 |
| 3 | Real Madrid | 0.28 | 0.28 | 0.31 |
| 4 | Athletic Bilbao | n.a. | 0.42 | 0.49 |
| 5 | Sevilla | 0.34 | 0.37 | 0.41 |
| 6 | Villarreal | 0.53 | 0.43 | 0.63 |
| 7 | Real Sociedad | 0.32 | 0.56 | 0.57 |
| 8 | Valencia | 0.38 | 0.52 | 0.35 |
| 9 | Celta | n.a. | 0.46 | 0.67 |
| 10 | Málaga | 0.45 | 0.63 | 0.38 |
| 11 | Espanyol | 0.39 | 0.55 | 0.46 |
| 12 | Eibar | 0.47 | 0.81 | 0.80 |
| 13 | La Coruña | 0.30 | 0.60 | 0.38 |
| 14 | Osasuna | 0.55 | 0.74 | 0.72 |
| 15 | Sporting Gijón | 0.28 | 0.63 | 0.40 |
| 16 | UD Las Palmas | 0.27 | 0.53 | 0.36 |
| 17 | Leganés | n.a. | n.a. | 0.76 |
| 18 | Levante | 0.62 | 0.48 | 0.75 |
| 19 | Mallorca | 0.56 | 0.45 | 0.63 |
| 20 | Zaragoza | 0.34 | 0.35 | 0.34 |
| 21 | Valladolid | 0.63 | 0.52 | 0.68 |
| 22 | Tenerife | 0.28 | 0.34 | 0.38 |
| 23 | Nastic Tarragona | n.a. | 0.54 | 0.45 |
| 24 | Huesca | n.a. | 0.71 | 0.59 |
| 25 | Albacete | n.a. | 0.34 | 0.54 |

Source: Author's elaboration.

The average value of the H-index in the sample of clubs for the 2016–2017 season is 0.51 and ranges from 0.28 for Real Madrid showing an almost homogeneous distribution of revenues among the four sources of funding to 0.81 for Eibar indicating an almost absolute dependence on broadcasting revenues.

### 4.2. Situation 2019–2020 Season: COVID Effect

The 2019–20 season was suspended on 12 March 2020 with 11 matchdays remaining, which were played without spectators in June and July of 2020. As a result, most of the season went on as normal, and the main economic consequence affected the revenue linked to the matchday. During the 2020–21 season, the consequences were even more evident.

In any case, after the first months of the pandemic, in the 2019/20 season, some financial imbalances occurred. As can be seen in Table 2, there were no significant changes in liquidity, solvency and indebtedness, continuing the trend initiated in previous years. The indicator that shows the worst performance in this season is economic profitability, which decreases, on average, from 10.4% in the 2016/17 season to 6.6% in the 2019/20 season. As can be seen in Table 1, despite the fact that this season's revenues grew compared to the 2016/17 season, the profit before interest and taxes plummeted by more than 80% and the losses of some clubs such as Málaga C.F. exceeded 20%. In other cases, some clubs went from showing profits to making losses. For instance, F.C. Barcelona club went from a 4% profit to a 7% loss, showing that the previous financial strength was deceptive. The problem with football finances is that although clubs are able to generate abundant and

largely predictable income, they are not able to retain that added value, which mostly goes to the actor who has the "irreplicable asset", i.e., the footballers.

*4.3. Influence of the Financial Variables and the Economic Control Regulation on Financial Dependence. An Econometrics Approach*

One of the main contributions of this work is to analyse the degree of financial dependence of the clubs through the H-index. So, it is interesting to carry out an analysis that measures the impact of these financial variables and the financial control system (FCS) implemented by the Spanish Professional Football League (SPFL) in 2015 on the financial dependency of clubs.

In the estimation reported in Table 5, we computed robust standard errors. Although we report the original coefficients, it is crucial to mention that they are not very useful for interpreting the results. For that reason, we also report their margins of response for conditional means in the next column. They are the average marginal effects and measure the amount of change in the H-index that is produced by a marginal change in each dependent variable[5].

**Table 5.** Estimations of Equation (2). Dependent variable H-index.

| Variables | BRM | | Margins BRM | |
|---|---|---|---|---|
| | Coef. | *p*-Value | Coef. | *p*-Value |
| Liquidity | 0.3284 | 0.019 | 0.1064 | 0.019 |
| Solvency | 0.0849 | 0.170 | 0.0275 | 0.170 |
| Indebtedness | 0.0011 | 0.642 | 0.0004 | 0.642 |
| ROA | 0.8196 | 0.008 | 0.2656 | 0.007 |
| Revenues | −0.0009 | 0.000 | −0.0003 | 0.000 |
| F. Control | 0.2886 | 0.002 | 0.0921 | 0.002 |
| Constant | −0.9781 | 0.000 | — | — |

The results reported in Table 5 indicate that all the variables, except Solvency and Indebtedness, are significant at standard levels. The relevant coefficients to read the results are those reported by Margins BRM. For instance, what the margin coefficient for Liquidity says is that when it raises one unit, the Herfindahl index increases by 10 per cent and each 1 per cent of change in ROA produces an H-index increase of 26.56%.

In addition, we can read the sign of each variable, then we can confirm that both Liquidity and ROA have a positive effect on the H-index, which means that these variables increase financial dependency. The only variable that reduces the dependency is Revenues. Another interesting finding is that the FCS implemented by the SPFL in 2015 increased the financial dependence of the clubs by about 9%.

## 5. Discussion

The results of the study seem to confirm that the different regulations implemented by both the State and the football governing bodies (UEFA and LFP) are contributing to stabilising the financial situation of Spanish professional football, as indicated by the CSD reports (CSD 2020) and recent literature (Calahorro-López et al. 2022; Urdaneta et al. 2021), beyond the limitations of accounting information derived from the application of accounting principles and rules that prevent the recognition as an asset of players for whom no transfer has been paid (de Liébana 2015; Morris 2013).

However, these improvements have been reflected in the solvency of the clubs rather than in their profitability. In addition, the analysis also warns about the large differences between clubs leading to a two-speed competition, and a possible shift from the traditional Utility Maximisation model (UM) to the Profit Maximisation model (PM) to ensure a better financial situation confirming previous forecasts by some authors (Nicoliello and Zampatti

2016). Though, it will be necessary to be careful with this type of affirmation, since the present study shows evidence that these improvements may increase the dependence on the same source of revenue that was finally spent on new signings that reinforce the capacity of the clubs for competitive results.

Real Madrid and F.C. Barcelona, the two most representative clubs in Spanish football, are examples of the management strategies mentioned above. Thus, Real Madrid seems to seek the sustainability of the club through financial viability at the time of trying to maximise sporting results, while F.C. Barcelona seeks the virtuous circle from sporting results that serve as an impulse to obtain financial resources. In this way, we can see that over the last decade Real Madrid's financial situation has shown some stability, even though it only won one league championship (although in this period it won four Champions League titles) during the 2013–2014 to 2018–2019 seasons. On the other hand, F.C. Barcelona has shown more leveraged financial ratios, but it won four leagues during the same period. However, its financial situation plummeted in 2019–2020 when it stopped winning titles. This strategy of FC Barcelona has been fundamentally materialised in expensive signings and costly renewals for long periods for veteran players whilst Real Madrid has a policy of annual renewals for players over 30 years, assuming the risk that some stars change teams, such as Ronaldo or Sergio Ramos.

However, the reality of these clubs is exceptional compared to the majority. Looking at all the clubs in the sample, the proportion between player wages and total income of the teams amounted to 73.01% (see Table 3) for the 2013/14 season, and 8 of 25 clubs exceeded 70% (see Appendix B), despite both the UEFA's FFP in its article 62 and the LIGA's RCE in its article 22 warning of a possible situation of economic and financial imbalance for those teams whose player wages exceeded 70% of their income. For the 2016/17 season, the clubs made efforts to improve this situation, and the sample's average dropped to 54.79%, and the number of teams with a percentage higher than 70% also was reduced to two clubs. However, for 2019/20, the average ratio rises again to 63.04% (see Table 3), as well as the number of teams to eight (see Appendix B).

In this sense, as stated by Nicoliello and Zampatti 2016, the control of expenditure and more specifically of the salaries of the sports staff is the key adjustment variable to control the financial result of football clubs. In this regard, the present study considers sports salaries to confirm more specifically what previous studies assumed about the ratio required by the regulations between personnel expenses and club income (Calahorro-López et al. 2022; Dimitropoulos and Scafarto 2021). To this end, the clubs try to attract talented players by paying high transfers and salaries for them as a strategy to shorten the maturity period of sporting success as much as possible (Barajas and Rodríguez 2010; Szymanski and Kuypers 1999).

However, the pure logic of competition shows that not all clubs can achieve their sporting objectives. Therefore, the initial financial outlay is still a gamble that, when it goes wrong, causes significant financial losses (Hamil et al. 2004).

Sometimes, foreign investment groups or eccentric millionaires, who, attracted by the popularity associated with sport, enter into the capital of clubs and present exciting sporting projects that they then abandon, leaving the club with the payment commitments acquired. This could be the case of Málaga C.F. or Racing Santander. The creation of the LFP to control clubs' spending in order to make them financially sustainable and avoid the temptation to overspend seems to be paying off (Nicoliello and Zampatti 2016).

When revenues are insufficient, either because of poor sporting performance or simply because the level of expenditure assumed was too ambitious, clubs are forced to transfer players to balance the books and comply with LFP's regulations. In the short term, this strategy allows clubs to improve their liquidity and solvency ratios, as we have seen in the previous section. Selling a player transforms an asset into a current asset that was either part of the entity's intangible fixed assets or was not even accounted as an asset if it was a junior player or acquired at zero cost.

However, in the medium and long term, this continuous decapitalisation leads to a gradual weakening of the institution and can initiate a vicious circle as described by Andreff (2018) for the French Ligue 1, in which a club's inability to retain the best players leads to poorer sporting results and, as a consequence, lower revenues, forcing the institution to sell players again, go into debt or increase capital to adjust the budget gaps. This situation creates a gap in competition on both sporting and economic levels.

A few clubs such as Real Madrid, Barcelona and more recently Atlético de Madrid have traditionally enjoyed a balanced, though risky, financial situation that has allowed them to compete systematically for titles, while the rest are often exposed to strong ups and downs. This is the case for Deportivo de la Coruña and Real Zaragoza, which have gone from winning national titles and competing in European competitions in the first decade of the 21st century to experiencing serious economic and sporting difficulties after relegation to second division.

Our analysis has shown that Spanish football is currently profitable for most clubs, with an average economic profitability of over 6% and only two clubs in the sample making losses. The reasons for this good performance of Spanish professional football, which contrasts with the widespread losses of clubs in Italian Calcio (Rey and Santelli 2017) and French Ligue 1 (Andreff 2018), are to be found in the cost controls imposed by the LFP's RCE (something that also happens in France and Germany) and, fundamentally, by the significant increase in broadcasting revenues in the Spanish league following the centralised sale of TV rights, which has particularly benefited the smaller clubs with less individual bargaining power. Thereby, television, which is football's main economic engine, is also its main risk due to the excessive financial dependence of most clubs on this source of income (see Appendix B), mainly because of the new broadcasting deal. An example of this can be seen in the season 2019–2020, where TV revenues represent more than 50% of the income for 17 of the 25 clubs of the sample.

A decade ago, Szymanski (2010) pointed out that in Spain only Real Madrid and Barcelona had real financial muscle, whilst the rest of the clubs struggle to compete on the basis of significant debt exposure. Recent events have shown that even the financial strength of these two clubs, in particular, F.C. Barcelona, was not as robust as it seemed. One of the great peculiarities of the football business, compared with other industries, is that the monopolistic position does not maximise profit, but clubs need competition to survive. Therefore, a scenario in which the ranking of the championships is practically predefined from the outset, as most teams systematically have to sell some of their best players to save their accounts, makes the product less attractive, and therefore increases the risk that future broadcasting contract negotiations will also be downward.

The results of the regression model show that improvements in liquidity and ROA have a positive impact on the H-index, generating the opposite effect than expected. This suggests that as clubs improve their ability to honour short-term commitments and their profitability, they maintain their focus on the main source of income (TV right), instead of diversifying to reduce this dependence.

The only variable whose coefficient points to a reduction in the H-index is Total Revenues, suggesting that the increase tends to be diluted among the different sources of income, and these are not used to improve the financial situation of the clubs, but possibly reinvested to reinforce the sporting competitiveness of the club.

Additionally, since the implementation of the UEFA FFP regulation, the H-index of the clubs has increased by around 9%. However, this is more likely to be a consequence of the centralisation of the sale of broadcasting rights implemented by Real Decreto-ley 5 (2015).

Finally, it must be added the uncertainty generated by the current COVID-19 pandemic. The loss of income due to the lack of spectators and the associated commercial income losses may cause a bursting of the possible bubble in transfer prices and salaries of footballers that have been systematically fed back with each signing of a shining star.

## 6. Conclusions

Spanish football has stood out over the last two decades for its sporting success both at the national team level and in international club competitions, and for its ability to attract some of the leading football stars of the moment, but at the same time, it has also gone through a serious financial crisis that has brought a good number of clubs to the brink of bankruptcy. This was probably the result of a management guided by sentimental or opportunistic issues of "fan" leaders, instead of applying the professional criteria required to manage such large budgets. Throughout this paper, we have analysed the effect of the implementation of the LFP's RCE of 2014, which was intended to improve the governance and financial sustainability of Spanish football clubs.

The financial indicators improved in most of the clubs in the sample, namely, 56% of the clubs increase their liquidity, 80% improve their solvency and 76% decrease their debt levels between the 2013/14 and 2016/17 seasons as a consequence of the new regulations (see Appendix A). However, these improvements are still slow to be seen in ROA, where less than half of the clubs (48%) increased this ratio.

It is precisely in ROA where the greatest impact of COVID-19 is observed, as the performance of 68% of the clubs was worse compared to the previous seasons, but as long as a number of clubs improve their liquidity, the solvency and debt ratios tend to remain the same. Therefore, while the regulation boosted the financial performance of clubs, the COVID-19 pandemic has slowed down this process, an effect that will become clearer in subsequent seasons.

Although the application of general accounting principles limits a proper interpretation and comparison of the financial statements of football clubs due to the assessment criteria of the sports squad, it is clear that the implementation of the RCE of clubs and SADs has allowed the financial situation of most clubs to be cleaned up and Spanish football to become a profitable industry. This has been helped not only by the control of expenditure brought about by the regulation but also by the significant increase in revenue as a result of the centralised sale of broadcasting rights by the LFP, which has benefited most of the smaller clubs, introducing certain solidarity in the distribution criteria. Beyond the momentary improvement of the financial situation, the RCE has not succeeded in transforming the "utility maximisation" (UM) approach into a "profit maximisation" (PM) strategy for Spanish football clubs.

This financial stability could only be transitory due to the extreme dependence of most clubs on TV funding, which has been highlighted by this study in terms of income concentration as a conditioning factor for the financial sustainability of clubs, causing large fluctuations in the income of the teams when sporting results are not as expected. This forces teams to transfer some of their best players to meet the commitment of financial balance. This practice pushes towards a loss of competitiveness in the league, as the best players are concentrated in a few teams, which detracts from the interest of the championship and could have consequences in future negotiations for the sale of rights. For all these reasons, for some years now, the creation of a European NBA-style "super league" has been discussed among the big clubs as a strategy to safeguard their financial interests.

Future lines could delve deeper into the impact of the regulations on the quality of accounting information in the Spanish league, as there is evidence of the introduction of earning management practices as a consequence of the new regulations (Dimitropoulos et al. 2016). In addition, the study presents a state of the art of Spanish football finances given the recent replacement of the UEFA FFP by the new UEFA Club Licensing and Financial Sustainability Regulations (UEFA 2022) implemented in June 2022 for the 2022/23 season. Therefore, in addition to the pending task of looking deeper into the impact of COVID-19 on football finances, a look at the repercussions of the new UEFA regulation, and perhaps more importantly, at the ongoing war in Ukraine, is added to the task at hand.

**Author Contributions:** Conceptualization, E.M.V. and R.U.-C.; formal analysis R.U.-C., J.C.G.-P. and N.L.C.; investigation, R.U.-C. and J.C.G.-P.; methodology, R.U.-C., J.C.G.-P., E.M.V. and N.L.C.; supervision, J.C.G.-P., E.M.V. and N.L.C.; validation, J.C.G.-P., E.M.V. and N.L.C.; writing—original draft, R.U.-C. and J.C.G.-P.; writing—review and editing, R.U.-C., J.C.G.-P., E.M.V. and N.L.C. All authors have read and agreed to the published version of the manuscript.

**Funding:** This research was funded by the Spanish Government Project PID2020-113905GB-I00 and the Project S56_20R, funded by the Aragon Regional Government.

**Data Availability Statement:** The datasets used during the current study are available from the corresponding author upon reasonable request.

**Acknowledgments:** We would like to appreciate the thoughtful and constructive advice provided by the reviewers, and especially the support of Héctor Jesús Millán Betancourt, external advisor to the Directorate of Statistics and Census of the Province of Río Negro and external advisor to the Under-Secretariat of Mining Development of Argentina.

**Conflicts of Interest:** The authors declare no conflict of interest.

**Appendix A. Financial Ratios Football Teams**

| ID | Teams | Liquidity | | | Solvency | | | Debt | | | ROA | | |
|---|---|---|---|---|---|---|---|---|---|---|---|---|---|
| | | 13–14 | 16–17 | 19–20 | 13–14 | 16–17 | 19–20 | 13–14 | 16–17 | 19–20 | 13–14 | 16–17 | 19–20 |
| 1 | Atlético Madrid | 0.6010 | 0.5567 | 0.5309 | 1.0452 | 1.0366 | 1.0988 | 22.1329 | 27.2999 | 10.1234 | 0.0344 | 0.0316 | 0.0428 |
| 2 | Barcelona | 0.3821 | 0.3572 | 0.3798 | 1.1216 | 1.1789 | 1.0245 | 8.2224 | 5.5905 | 40.8912 | 0.1175 | 0.0408 | −0.0677 |
| 3 | Real Madrid | 0.7352 | 0.7063 | 0.7663 | 1.6075 | 1.7583 | 1.5914 | 1.6462 | 1.3187 | 1.6908 | 0.0443 | 0.0257 | 0.0003 |
| 4 | Athletic Bilbao | 1.1041 | 1.6292 | 2.6845 | 2.2062 | 2.6772 | 1.8007 | 0.8290 | 0.5962 | 1.2489 | 0.2545 | 0.1314 | −0.0653 |
| 5 | Sevilla | 0.3927 | 0.8095 | 0.7828 | 1.4560 | 1.5946 | 1.4057 | 2.1928 | 1.6819 | 2.4651 | 0.0747 | 0.1489 | 0.0082 |
| 6 | Villarreal | 1.5057 | 0.8329 | 1.0092 | 4.0779 | 1.6549 | 1.7802 | 0.3249 | 1.5269 | 1.2818 | 0.0718 | −0.0159 | −0.0001 |
| 7 | Real Sociedad | 1.0618 | 0.6698 | 0.3738 | 1.3001 | 1.8692 | 1.7772 | 3.3320 | 1.1505 | 1.2866 | 0.3091 | 0.0213 | 0.0183 |
| 8 | Valencia | 0.1836 | 0.4269 | 0.2131 | 1.1383 | 1.2240 | 1.0929 | 7.2288 | 4.4645 | 10.7598 | 0.0305 | 0.0027 | −0.0157 |
| 9 | Celta | 0.8803 | 1.5441 | 1.5530 | 1.2921 | 2.3025 | 2.3446 | 3.4230 | 0.7677 | 0.7437 | 0.3646 | 0.3004 | 0.0855 |
| 10 | Málaga | 0.2683 | 0.5323 | 0.8049 | 0.5095 | 1.3325 | 1.6342 | −2.0386 | 3.0077 | 1.5767 | 0.3448 | 0.0507 | −0.2027 |
| 11 | Espanyol | 0.6146 | 0.6051 | 0.3293 | 1.0737 | 1.2295 | 1.9056 | 13.5593 | 4.3565 | 1.1042 | 0.0169 | 0.0313 | 0.0678 |
| 12 | Eibar | 1.1249 | 1.5267 | 1.3448 | 1.6049 | 1.9220 | 3.7841 | 1.6532 | 1.0845 | 0.3592 | 0.0788 | 0.2333 | 0.1421 |
| 13 | La Coruña | 0.1828 | 0.6467 | 0.1449 | 0.1756 | 0.4010 | 0.2585 | −1.2129 | −1.6695 | −1.3486 | −0.1458 | 0.5026 | 0.0868 |
| 14 | Osasuna | 0.7029 | 0.7664 | 0.3099 | 1.0415 | 1.4457 | 1.4361 | 24.1096 | 2.2435 | 2.2929 | −0.0148 | 0.4110 | 0.0432 |
| 15 | Sporting Gijón | 0.3178 | 0.2794 | 0.3726 | 0.5566 | 1.6328 | 1.9659 | −2.2552 | 1.5802 | 1.0353 | 0.0521 | 0.1687 | 0.0038 |
| 16 | UD Las Palmas | 0.3966 | 0.7898 | 1.1875 | 1.1983 | 1.5163 | 2.1938 | 5.0422 | 1.9369 | 0.8377 | 0.0318 | 0.2102 | 0.0301 |
| 17 | Leganés | 1.0469 | 1.1174 | 4.1302 | 6.9034 | 1.1726 | 4.0993 | 0.1694 | 5.7934 | 0.3227 | −0.0001 | 0.1089 | 0.3760 |
| 18 | Levante | 0.8659 | 0.6615 | 0.5226 | 1.2359 | 1.4655 | 1.5537 | 4.2386 | 2.1480 | 1.8060 | 0.0751 | 0.0521 | 0.0126 |
| 19 | Mallorca | 0.7940 | 0.8339 | 1.2819 | 1.0439 | 1.2461 | 1.8657 | 22.7878 | 4.0631 | 1.1552 | −0.0985 | −0.1089 | 0.3593 |
| 20 | Zaragoza | 0.1996 | 0.2724 | 0.4257 | 0.9473 | 1.0167 | 1.1463 | −18.9682 | 59.8743 | 6.8332 | −0.0279 | 0.0307 | 0.0920 |
| 21 | Valladolid | 1.8686 | 0.4556 | 0.3828 | 0.5739 | 1.0132 | 1.0544 | −2.3471 | 75.8422 | 18.3694 | 0.1115 | 0.0068 | 0.2855 |
| 22 | Tenerife | 0.2963 | 0.6914 | 0.6338 | 1.1414 | 1.4729 | 1.7902 | 7.0704 | 2.1145 | 1.2654 | 0.1124 | 0.1669 | 0.0598 |
| 23 | Nastic Tarragona | 0.6079 | 0.9278 | 1.4357 | 1.2189 | 1.5393 | 1.9314 | 4.5677 | 1.8543 | 1.0736 | 0.0361 | 0.1502 | 0.0061 |
| 24 | Huesca | 3.9567 | 1.2616 | 1.3324 | 4.0209 | 2.3203 | 2.4713 | 0.3310 | 0.7574 | 0.6797 | 0.0324 | 0.0361 | 0.1926 |
| 25 | Albacete | 0.2315 | 0.1147 | 1.0174 | 1.0252 | 0.9295 | 1.2078 | 39.6317 | −14.1863 | 4.8122 | 0.1378 | −0.1292 | 0.1011 |

**Appendix B. Main Incomes and Player Wages in Proportion to Total Revenues**

| ID | Teams | 2013–2014 | | | | | 2016–2017 | | | | | 2019–2020 | | | | |
|----|-------|----|--------|---------|------|-----------|----|--------|---------|------|-----------|----|--------|---------|------|-----------|
| | | TV | Market | Tickets | O. I. | P. Wages | TV | Market | Tickets | O. I. | P. Wages | TV | Market | Tickets | O. I. | P. Wages |
| 1 | Atl. Madrid | 28% | 19% | 11% | 42% | 63% | 38% | 21% | 10% | 31% | 61% | 35% | 22% | 11% | 32% | 58% |
| 2 | Barcelona | 37% | 37% | 12% | 15% | 49% | 31% | 45% | 9% | 16% | 58% | 35% | 46% | 8% | 11% | 61% |
| 3 | R. Madrid | 33% | 28% | 28% | 12% | 48% | 25% | 38% | 25% | 13% | 54% | 21% | 45% | 18% | 15% | 59% |
| 4 | Ath. Bilbao | n.a. | n.a. | n.a. | n.a. | 77% | 58% | 4% | 25% | 13% | 53% | 66% | 5% | 22% | 7% | 93% |
| 5 | Sevilla | 47% | 15% | 9% | 28% | 75% | 52% | 10% | 10% | 28% | 65% | 57% | 12% | 6% | 25% | 74% |
| 6 | Villarreal | 70% | 17% | 7% | 5% | 55% | 60% | 15% | 4% | 21% | 60% | 77% | 18% | 3% | 1% | 87% |
| 7 | R. Sociedad | 40% | 10% | 13% | 37% | 62% | 73% | 14% | 10% | 3% | 60% | 74% | 14% | 10% | 2% | 75% |
| 8 | Valencia | 56% | 16% | 13% | 16% | 58% | 69% | 12% | 12% | 6% | 70% | 44% | 12% | 7% | 36% | 58% |
| 9 | Celta | n.a. | n.a. | n.a. | n.a. | 45% | 64% | 8% | 7% | 22% | 50% | 81% | 13% | 5% | 1% | 68% |
| 10 | Málaga | 63% | 12% | 18% | 8% | 41% | 78% | 12% | 7% | 4% | 49% | 52% | 29% | 14% | 4% | 83% |
| 11 | Espanyol | 54% | 22% | 22% | 2% | 46% | 72% | 14% | 14% | 1% | 62% | 64% | 14% | 8% | 15% | 64% |
| 12 | Eibar | 64% | 20% | 10% | 5% | 43% | 90% | 7% | 2% | 1% | 55% | 89% | 8% | 2% | 1% | 64% |
| 13 | La Coruňa | 32% | 18% | 38% | 11% | 68% | 76% | 11% | 10% | 3% | 44% | 52% | 17% | 27% | 4% | 77% |
| 14 | Osasuna | 72% | 12% | 14% | 2% | 73% | 85% | 6% | 7% | 1% | 25% | 84% | 10% | 5% | 1% | 54% |
| 15 | Sporting | 31% | 25% | 34% | 10% | 65% | 78% | 8% | 10% | 4% | 36% | 53% | 20% | 26% | 0% | 51% |
| 16 | Las Palmas | 36% | 21% | 19% | 23% | 63% | 71% | 11% | 13% | 6% | 34% | 44% | 39% | 10% | 8% | 52% |
| 17 | Leganés | n.a. | n.a. | n.a. | n.a. | 96% | n.a. | n.a. | n.a. | n.a. | 30% | 87% | 9% | 4% | 1% | 50% |
| 18 | Levante | 78% | 14% | 6% | 2% | 59% | 64% | 24% | 10% | 1% | 105% | 86% | 10% | 2% | 2% | 52% |
| 19 | Mallorca | 72% | 18% | 8% | 2% | 81% | 60% | 27% | 10% | 2% | 55% | 78% | 15% | 5% | 2% | 36% |
| 20 | Zaragoza | 46% | 20% | 28% | 5% | 108% | 50% | 23% | 23% | 5% | 40% | 48% | 17% | 26% | 8% | 47% |
| 21 | Valladolid | 78% | 6% | 16% | 0% | 67% | 69% | 16% | 15% | 0% | 56% | 81% | 11% | 8% | 0% | 44% |
| 22 | Tenerife | 37% | 28% | 22% | 12% | 48% | 46% | 31% | 11% | 12% | 38% | 48% | 37% | 10% | 6% | 51% |
| 23 | Nastic T. | n.a. | n.a. | n.a. | n.a. | 124% | 71% | 13% | 14% | 2% | 46% | 3% | 46% | 49% | 2% | 81% |
| 24 | Huesca | n.a. | n.a. | n.a. | n.a. | 64% | 84% | 7% | 6% | 4% | 61% | 75% | 14% | 10% | 1% | 84% |
| 25 | Albacete | n.a. | n.a. | n.a. | n.a. | 245% | 3% | 46% | 26% | 25% | 101% | 70% | 18% | 11% | 1% | 53% |

Note: TV: TV rights between total income, Market: Advertising and Marketing income between total income, Tickets: Match Day income between total income, O. I.: Other Income between total income and P. Wages: Player wages between total income.

## Notes

[1]  However, it is very difficult to isolate the effect of the entry into force of the economic control measures from the rest of the changes that occurred in the period. The relationship established in the econometric model and the results should be considered with caution since it is a field that requires more theoretical and empirical work, in order to unravel the true causal relationships between the variables.

[2]  The latest contract refers to the sale of the broadcasting rights for the 2022–23 to 2026–27 first-division seasons for EUR 4.95 billion.

[3]  The 4 clubs that have not signed the agreement with CVC are Real Madrid, F.C. Barcelona, Ath. Bilbao and Oviedo.

[4]  We have also considered the fractional regression model proposed by Papke and Wooldridge (1996, 2008), obtaining the same results.

[5]  In other words, average marginal effects are d(y)/d(x) for continuous variables, and Δy/Δx for dummy variables x. To obtain a better understanding of margin coefficients, we recommend Cameron and Trivedi (2010) and Williams (2012).

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
