# Peer review of "The Other Side of the “League of Stars”: Analysis of the Financial Situation of Spanish Football"

_ijfs, doi:10.3390/ijfs11010003_

Round 1
Reviewer 1 Report
1. There is a certain gap between the objective of the study and the research carried out. The introduction states that the objective of the study is "to try to assess the viability of the current management model of Spanish football". However, the paper does not present any management model of football and therefore it is not possible to assess it. In contrast, the paper analyses changes in the financial situation of Spanish football clubs before and after the implementation of the financial control system in 2015, including an assessment of the results of the Covid-19 period. Therefore, I consider that the objective of the paper should be clarified by linking it more closely to the results of the research carried out, i.e. the evaluation of the changes in the financial situation of clubs between 2013 and 2020.
2. The research methods chosen for the study are appropriate, but a clearer rationale for the choice of these four financial indicators is needed. For example, the most popular indicator in the profitability group is return on sales, but the study uses return on assets (ROA). It would be useful to explain why this indicator is more appropriate for analysing the financial situation of football clubs. Also, why an EBIT base is chosen for the calculation of ROA, whereas the normal base is net income. Perhaps this is specific to this activity, but it would be useful to comment on it. The same applies to the other indicators. For example, there are several liquidity ratios, but the authors have chosen to use a current ratio. Why not a quick liquidity ratio? The quick liquidity ratio gives an even clearer indication of the level of liquidity. An explanation is also needed.
3. Conclusions lack specificity. It is recommended that the conclusions should highlight more clearly the results of the study, indicating to what extent the financial indicators analysed have improved as a result of the introduction of the financial control system and to what extent they have been influenced by Covid-19.
3. Conclusions lack specificity. It is recommended that the conclusions should highlight more clearly the results of the study, indicating to what extent the financial indicators analysed have improved as a result of the implementation of the financial control system and to what extent they have been influenced by Covid-19.
Author Response
Dear reviewer,
We attend your observations in the attached file.
Kind regards,
The authors

Author Response
Dear reviewer,
Please see the attachment (please do not take into account the file author-coverletter-23023951 as it has been uploaded by mistake).
Regards,
The authors

Reviewer 3 Report
1. Research Objectives
Page 2, Lines 64-67,
This manuscript should explain how the current methodology could show the dynamic analysis of the financial situation of Spanish clubs
2. Methodology
This paper has the unique team-level Herfindahl index, financial ratios, income distribution data in Table 4, Appendix 1 and 2. However, the current manuscript only used the descriptive statistics to generate findings. The referee suggests that this paper is required to examine the relationship between Herfindahl index, financial ratios and income distribution in order to highlight the contribution of this paper.
3. Quality of this paper:
This manuscript should pay attention on these typos carefully as below,
Page 4, Table 1 Typo “Media”. It should be Median
Page 6, Table 3 Typo “Macth day”. It should be “Match day”.
Author Response

(The authors gave the same response as above.)

Round 2
Reviewer 2 Report
The paper is valid in its present form
Author Response
Thanks very much
Reviewer 3 Report
The contribution of this manuscript is very limited if this manuscript only used the descriptive statistics to demonstrate their findings. Even though this revised manuscript further used the correlation analysis without showing the significance level, it is not enough to support the causal relations between HI and financial variables. Of particular, the reviewer suggests that this manuscript should use the panel regression to demonstrate the causal relationship by different periods.
Author Response
Dear Reviewer,
thanks you very much for your comments.
In the new version we have modified the section 3 (methodology) to explain the
rationale behind the panel data econometric analyses we have used, adding a
subheading in the results section (4.3) that explains the regression results. Therefore, we have removed the analysis of correlations of the Herfindahl index with financial ratios and total revenues.
We consider that this has substantially improved although the relationship established in the econometric model and the results should be considered with caution since it is a field that requires more theoretical and empirical work, in order to unravel the true causal relationships between the variables.
Yours sincerely